Interaction between crizotinib and tropifexor through in vitro and in vivo studies

Shen An 1
Xia Hailun 2
Wu Jun 2
Tao Liang 1
Chen Jie 2
Lin HangJuan nbszyy_lhj@126.com 1
1 Department of Pharmacy, Ningbo Municipal Hospital of Traditional Chinese Medicine (TCM), Affiliated Hospital of Zhejiang Chinese Medical University , Ningbo , Zhejiang , China
2 Department of Pharmacy, The First Affiliated Hospital of Wenzhou Medical University , Wenzhou , Zhejiang , China
Anson Lesley
Electronic publication date: 2025 Oct 22
Publication date: 2025
Volume: 13
Electronic Location ID: e20256
Received 2025 May 12; Accepted 2025 Sep 26
Copyright: ©2025 Shen et al.
Copyright year: 2025
Copyright holder: Shen et al.
License: This is an open access article distributed under the terms of the Creative Commons Attribution License, which permits unrestricted use, distribution, reproduction and adaptation in any medium and for any purpose provided that it is properly attributed. For attribution, the original author(s), title, publication source (PeerJ) and either DOI or URL of the article must be cited.
License URL: https://creativecommons.org/licenses/by/4.0/

Keywords: Crizotinib, Methodological, Verification, UPLC-MS/MS, Pharmacokinetics

Funding: Ningbo medical and health brand discipline project No. PPXK2024-08 Ningbo Traditional Chinese Medicine Pharmaceutical Preparation Center Construction Project Project No. zyy23011 This work was supported by Ningbo medical and health brand discipline project (No. PPXK2024-08), Ningbo Traditional Chinese Medicine Pharmaceutical Preparation Center Construction Project (Project No. zyy23011). The funders had no role in study design, data collection and analysis, decision to publish, or preparation of the manuscript.

==============================
In the context of cancer treatment, the employment of multiple drug therapies frequently results in a high prevalence of drug-drug interaction (DDI) in clinical practice. Crizotinib is a tyrosine kinase inhibitor (TKI) used to treat non-small cell lung cancer (NSCLC). Tropifexor is a Farnesoid X Receptor (FXR) agonist used to treat non-alcoholic steatohepatitis (NASH) and other metabolic disorders. This study developed an ultra performance liquid chromatography-tandem mass spectrometry (UPLC-MS/MS) method for the quantitative determination of crizotinib and 2-Keto crizotinib concentrations and investigated the effect of tropifexor on crizotinib metabolism. Results showed good linearity for crizotinib and 2-Keto crizotinib in plasma, with the method meeting all quantitative analysis requirements, including selectivity, accuracy, precision, stability, matrix effects, and recovery. In rat liver microsomes (RLM), tropifexor inhibited the metabolism of crizotinib via non-competitive and uncompetitive mechanisms, whereas in human liver microsomes (HLM), the inhibition occured through competitive and non-competitive mechanisms. In vivo studies in rats demonstrated that tropifexor significantly increased the AUC0−t, AUC0−∞, and Cmax of crizotinib by 35.7%, 36.9%, and 37.5%, respectively, and decreased the CLz/F of crizotinib by 25.2%. For the metabolite 2-Keto crizotinib, tropifexor reduced its CLz/F by 27.9%. Our study developed this UPLC-MS/MS method for the accurate and sensitive quantitative determination of crizotinib and 2-Keto crizotinib concentrations, and elucidated the inhibitory effect of tropifexor on crizotinib metabolism and its inhibitory mechanism. The results of this study will support the necessity of monitoring crizotinib plasma concentrations when used in combination therapy.

Introduction

According to GLOBOCAN, lung cancer was the most common cancer worldwide in 2022, accounting for 12.4% of all new cancer cases (approximately 2.5 million cases) and becoming the leading cause of cancer-related deaths globally (accounting for 18.7%) (Bray et al., 2024). Non-small cell lung cancer (NSCLC), accounting for 85% of malignant lung tumors, is one of the most prevalent cancer types and remains a leading cause of cancer-related mortality worldwide (Hendriks et al., 2024). The World Health Organization recognizes the significant impact of lung cancer on global health and has implemented several initiatives to comprehensively address the disease. In recent years, driven by breakthroughs in targeted therapy and immunotherapy, the field of NSCLC management has experienced exponential growth (Jeon et al., 2025). Clinically, tyrosine kinase inhibitor (TKI) has emerged as a preferred first-line treatment option in cancer therapy (He et al., 2021). Crizotinib, a small-molecule TKI, is the first member of the anaplastic lymphoma kinase inhibitor (ALKi) family (D’Angelo et al., 2020). In humans, crizotinib is metabolized by CYP3A4 primarily in the liver to form 2-Keto crizotinib, with approximately 63% of the drug being excreted unchanged in the feces (Johnson et al., 2015). CYP3A4 is a critical enzyme in the metabolism of pharmaceuticals, and its activity is a primary contributor to the occurrence of drug-drug interaction (DDI).

The antitumor efficacy of crizotinib has been demonstrated in multiple clinical trials, showing significant therapeutic benefits, particularly in ALK-positive NSCLC (Shaw et al., 2020; Shaw et al., 2019). Common adverse effects during crizotinib treatment include gastrointestinal discomfort, visual disturbances, and, most notably, hepatotoxicity (Van Geel et al., 2016; Wang et al., 2019; Das & Das, 2021). In one study, 17% of patients receiving the standard dose of crizotinib experienced elevated alanine aminotransferase (ALT) levels (Kazandjian et al., 2014). Additionally, research indicates that patients with pre-existing liver conditions are 2.3 times more likely to experience crizotinib-induced hepatotoxicity than those without liver disease (Jung et al., 2018). Considering the hepatotoxic potential of crizotinib and the high prevalence of comorbidities among cancer patients, combination therapies are often utilized in clinical practice (Moosavi et al., 2021). However, there is a potential for DDI to occur with combination medications, leading to fluctuations in drug exposure and affecting the therapeutic efficacy and safety of the drug. As a CYP3A4 substrate, crizotinib carries a high risk of increased plasma exposure when used in combination with CYP3A inhibitors, thereby increasing the risk of adverse reactions and affecting drug safety (Wang et al., 2024; Yamazaki, Johnson & Smith, 2015). In this study, we selected tropifexor as a potential inhibitor to investigate its effect on the metabolism of crizotinib and assess the risk of DDI.

Tropifexor is a novel and potent Farnesoid X Receptor (FXR) agonist that has demonstrated high efficacy in treating liver diseases, such as non-alcoholic steatohepatitis (NASH), by reducing liver inflammation and fibrosis (Chen et al., 2022a). In a liver injury model, tropifexor had been shown to improve hepatic transaminase levels and reduce fibrosis (Trauner et al., 2017). In a first-in-human study involving healthy volunteers, a single dose of up to three mg of tropifexor was found to be safe and well-tolerated (Badman et al., 2020). Furthermore, existing data suggest that in a cholestasis pig model, tropifexor effectively inhibits liver and brain damage induced by bile duct ligation (BDL) through improving glutamine metabolism and the urea cycle (Xiao et al., 2024). In light of the evident benefits associated with tropifexor, there is considerable promise for its integration within combination therapy regimens. However, the increased risk of DDI in combination therapies highlights the critical need for further research into their efficacy and safety (Li et al., 2023; Hurtado et al., 2021).

To the best of our knowledge, previous studies have constructed the bioanalytical methods to detect the concentration of crizotinib in plasma, without the metabolite 2-Keto crizotinib (Li, Zhao & Zhao, 2022; Hollander et al., 2024; Bellouard et al., 2023; Chen et al., 2023; Lou et al., 2022; Van Veelen et al., 2021; Maher et al., 2021; Zhou et al., 2021; Mukai et al., 2021; Zhao et al., 2020; Reis et al., 2018). To date, there was only one available LC-MS/MS approach to measure crizotinib and its metabolite in plasma together (Qi et al., 2018). However, this method had long analytical time (3.0 min). Consequently, a comprehensive ultra performance liquid chromatography tandem mass spectrometry (UPLC-MS/MS) method for the detection of crizotinib and its metabolites 2-Keto crizotinib in plasma is necessary to be constructed and validated.

Thus, in this study, we established and validated an UPLC-MS/MS detection method capable of accurately, rapidly, and sensitively detecting crizotinib and its metabolite 2-Keto crizotinib. Additionally, we utilized this detection method to investigate the inhibitory effect of tropifexor on crizotinib metabolism and its inhibitory mechanism in rat liver microsomes (RLM) and human liver microsomes (HLM). Finally, we used a Sprague-Dawley (SD) rat model to study the effects of tropifexor on crizotinib metabolism in vivo. We hope that our findings will provide data support for clinicians to adjust dosing regimens.

Materials & Methods

Chemicals and reagents

Crizotinib (CAS: 877399-52-5), 2-Keto crizotinib (CAS: 1415558-82-5), and alectinib (CAS: 1256580-46-7, used as internal standard, IS) were supplied from Beijing Sunflower Technology Development Co., Ltd (Beijing, China). Tropifexor (CAS: 1383816-29-2) was provided by Shanghai Canspec Scientific Instruments Co., Ltd. (Shanghai, China). The purity of the drugs used in the experiment was ≥98%. RLM was prepared following established protocols from the relevant literature (Wang et al., 2015), and the protein concentration was determined to be 35.00 mg/mL using the Bradford Protein Assay Kit (Thermo Fisher Scientific, Waltham, MA, USA). HLM was acquired from iPhase Pharmaceutical Services Co., Ltd (Beijing, China). Methanol and acetonitrile for chromatography were purchased from Merck (Darmstadt, Germany). Ultra-pure water for laboratory research was provided by a Milli-Q water purification system (Millipore, Bedford, MA, USA). All other chemicals and solvents not mentioned above were of analytical grade.

Equipment and operating conditions

The concentrations of crizotinib, the metabolite 2-Keto crizotinib and IS were detected by UPLC-MS/MS, which was equipped with a Waters Acquity UPLC BEH C18 column (2.1 mm × 50 mm, 1.7 µm particle size; Waters Corp., Milford, MA, USA). The column temperature was maintained at 40 °C, while the automatic sampler was kept at 4 °C. The mobile phase was composed of solution A (0.1% formic acid in water) and solution B (acetonitrile), with the gradient elution program set as follows over a 2.0 min running: 0–0.5 min, 90% A; 0.5–1.0 min, 90–10% A; 1.0–1.4 min, 10% A; 1.4–1.5 min, 10–90% A; 1.5–2.0 min, 90% A. The flow rate of the mobile phase was 0.4 mL/min. The multiple reaction monitoring (MRM) mode was used, and the chemical structures are depicted in Fig. 1. The instrument was operated in positive ion electrospray ionization (ESI) mode. The ion transitions for crizotinib, 2-keto crizotinib, and IS were 450.06 → 260.07, 463.96 → 274.02, and 482.99 → 395.99, respectively. And the cone voltages were 30, 30, and 20 V, respectively, and the collision energies were 20, 20, and 21 eV, respectively.

Figure 1 Mass spectrometric analysis and chemical formulas of crizotinib (A), 2-Keto crizotinib (B), and alectinib (IS, C).

Calibration standards and quality control

Standard stock solutions of crizotinib (1 mg/mL), 2-Keto crizotinib (1 mg/mL) and IS (1 mg/mL) were separately prepared by dissolving in methanol. The IS working solution (400 ng/mL) and the working solutions for calibration curves and quality control (QC) samples were prepared daily by diluting their respective stock solutions with methanol. The calibration standards were prepared by adding 10 µL crizotinib or 2-Keto crizotinib working solutions with different concentrations into 90 µL blank rat plasma. The final concentrations of calibration standards were 2, 5, 10, 20, 50, 100 and 300 ng/mL for crizotinib and 1, 2, 5, 10, 20, 50 and 100 ng/mL for 2-Keto crizotinib, respectively. The sensitivity of the method was reflected by the lower limit of quantification (LLOQ), which was also the lowest quantitative point. The LLOQ and three QC samples were obtained by the same method. The LLOQ of crizotinib and 2-Keto crizotinib were 2 ng/mL and 1 ng/mL, respectively, and the three QC samples of crizotinib and 2-Keto crizotinib were 5, 120, 240 ng/mL, and 2, 40, 80 ng/mL, respectively. All chemicals and solutions were stored at −80 °C for further use.

Pre-treatment of samples

Protein precipitation was used to remove proteins from plasma and extract the test materials. To this end, 100 µL of plasma sample was combined with 10 µL of the IS working solution (400 ng/mL) and 300 µL of acetonitrile. The mixture was then centrifuged at 13,000 rpm for 10 min at 4 °C to ensure complete protein precipitation. After centrifugation, the supernatant was carefully collected and transferred into a sample vial. Finally, 2.0 µL of sample was injected for UPLC-MS/MS analysis.

Method validation

The calibration curve, selectivity, accuracy, precision, recovery, matrix effect, and stability of the method established in this study were validated according to the FDA Bioanalytical Method Validation Guidance and the guidelines from the National Medical Products Administration (NMPA). Detailed information regarding the evaluation could be found in the Results and Discussion sections.

Establishment of enzyme incubation system in vitro

A 200 µL incubation system was used, which included 0.1 M phosphate buffer (PH 7.4), 0.3 mg/mL RLM or HLM, 1 mM NADPH, and 0.1–50 µM crizotinib. Before measurement, the mixture without NADPH was pre-incubated at 37 °C for 5 min. Then, one mM of NADPH was given to incubate the reaction in a water bath with shaking at a rate of 200 times/min. After 30 min, the reaction was terminated by cooling at −80 °C. Next, 300 µL acetonitrile and 10 µL of the IS working solution were added to the mixture. After shaking for 2 min, the supernatant was obtained by centrifugation at 13,000 rpm for 10 min and subjected to UPLC-MS/MS analysis. With this process, we obtained the Michaelis–Menten constant (Km) values of crizotinib in RLM and HLM.

To assess the inhibitory effect of tropifexor, a range of tropifexor concentrations (0, 0.01, 0.1, 1, 10, 25, 50, 100 µM) were tested in RLM and HLM using the corresponding Km as the concentration of crizotinib to determine the half-maximal inhibitory concentration (IC50). Subsequently, to determine the type of inhibition mechanism of tropifexor on crizotinib metabolism, based on the Km values, the concentrations of crizotinib were set (0.25, 1.00, 1.50, and 2.00 µM in RLM; 3.75, 7.50, 15.00, and 30.00 µM in HLM). In addition, the concentrations of tropifexor were adjusted according to the IC50 values, resulting in a range of 0, 1.08, 4.34, 8.67 µM in RLM, and 0, 1.08, 2.15, 4.30 µM in HLM, respectively. The subsequent reactions and processing steps were carried out as described above and analyzed using UPLC-MS/MS.

In vivo pharmacokinetic study

The study was approved by Institutional Animal Care and Use Committee of the First Affiliated Hospital of Wenzhou Medical University (Approval No. WYYY-IACUC-AEC-2024-079). The experimental animals were cared in accordance with the Guide for the Care and Use of Laboratory Animals issued by the National Research Council, complying with the ARRIVE guidelines. The 6–8 weeks old Sprague-Dawley rats were obtained from Vital River Laboratory Animal Technology Co., Ltd (China, Zhejiang). The rats were housed at 25 °C and given a 14-day acclimation period under experimental conditions to reduce potential variability.

Due to the similarity between their hepatic enzyme system and humans, SD male rats are widely used in pharmacokinetics (Riccardi et al., 2018; Martignoni, Groothuis & De Kanter, 2006). A total of 10 SD male rats (200 ± 20 g) were randomly divided into two groups (n = 5): Group A (control group) and Group B (experimental group). The animals were fasted for 12 h prior to the experiment but had free access to water. Crizotinib was prepared as a suspension in 0.5% carboxy methyl cellulose sodium (CMC-Na) solution, while tropifexor was dissolved in corn oil. The concentration of crizotinib was determined to be 25 mg/kg based on prior literature (Bland et al., 2020). And the concentration of tropifexor was determined to be 20 µg/kg based on body surface area conversion from human administered doses in prior literature (Chen et al., 2022a). Experimental group was received of tropifexor (20 µg/kg) via oral gavage, whereas control group was administered of an equivalent volume of corn oil. 30 min later, all rats were given crizotinib (25 mg/kg) by oral gavage. Blood samples were then collected at different time points at 0.5, 1, 1.5, 2, 4, 8, 12, 24, and 48 h after administration. Each blood sample (approximately 0.3 mL) was taken from the caudal veins, and was collected into 1.5 mL heparinized polyethylene tubes. After centrifuged at 13,000 rpm for 10 min, the supernatant was collected and stored at −80 °C. The plasma sample was treated according to 2.4 Pre-treatment of samples before UPLC-MS/MS analysis.

Animals were euthanized using the anesthesia method according to the AVMA Guidelines for Animal Euthanasia. All experimental animals were euthanized with intravenous pentobarbital (150 mg/kg) after completion of the experiment. After ensuring that the animals were free of life pointers, they were packaged and cremated.

Data analysis

The Km, IC50, the Lineweaver-Burk plot and the mean plasma concentration–time curve were calculated and plotted using GraphPad Prism 9.0 software (GraphPad Software Inc., California, United States). Pharmacokinetic parameters were calculated using the Drug and Statistics (DAS) software (version 3.0 software, Mathematical Pharmacology Professional Committee of China, Shanghai, China) with non-compartment model analyses based on the input data of drug administration route, dosage, drug concentration, and time. The comparison of the pharmacokinetic parameters were performed with SPSS (version 26.0; SPSS Inc., Armonk, NY, USA), with student t-test, and the P-value <0.05 was regarded as statistically significant.

Results

Method validation

The chromatograms in Fig. 2 showed that within 2.0 min of elution time, the retention times of crizotinib, 2-Keto crizotinib, and IS were 1.13 min, 1.17 min, and 1.21 min, respectively, achieving clear separation without interference from endogenous substances. Additionally, crizotinib and 2-Keto crizotinib exhibited good linearity over the concentration ranges of 2–300 ng/mL and 1–100 ng/mL, respectively. The regression equations for the calibration curves were as follows: crizotinib, y= (0.004216*x−0.001441, r2 = 0.995); 2-Keto crizotinib, y = (0.039363*x−0.004714, r2 = 0.993).

Figure 2 Representative MRM chromatograms of crizotinib, 2-Keto crizotinib (PF-0620182) and alectinib (IS) in rat plasma.

A blank rat plasma sample ((A); no analyte, no IS); Blank plasma sample with standard analytes and IS added (B); Plasma sample from rats received a single dose of 25 mg/kg crizotinib (C).

The LLOQ of crizotinib and its metabolite were 2 ng/mL and one ng/mL, respectively. Precision and accuracy were assessed using four concentration levels for each analyte. For crizotinib and 2-Keto crizotinib, both intra-day and inter-day precision were below 15%, and accuracy was within ±15% at the LLOQ and three QC levels. Detailed results are summarized in Table 1.

Table 1 The precision, accuracy, recovery and matrix effect of crizotinib and 2-Keto crizotinib in rat plasma (n = 5).

Analytes	Concentration
(ng/mL)	Intra-Day	Inter-Day	Recovery (%)	Matrix effect (%)	
		Precision
(RSD %)	Accuracy
(RE %)	Precision
(RSD %)	Accuracy
(RE %)	Mean ± SD	RSD%	Mean ± SD	RSD%	
Crizotinib	2	7.7	−3.2	8.0	−4.5					
	5	4.3	−1.4	3.7	−0.9	94.2 ± 5.3	5.6	93.3 ± 7.3	7.8	
	120	1.8	1.7	1.8	0.3	100.5 ± 6.4	6.4	94.8 ± 5.1	5.3	
	240	3.0	−1.2	3.5	−0.7	96.7 ± 3.2	3.3	100.2 ± 3.6	3.6	
2-Keto Crizotinib	1	3.3	3.2	4.1	1.1					
	2	2.8	2.2	3.0	1.0	92.5 ± 5.0	5.4	98.4 ± 5.1	5.2	
	40	1.8	−0.3	2.0	−1.0	93.5 ± 2.3	2.5	102.7 ± 2.5	2.4	
	80	2.8	−3.1	2.3	−3.1	92.9 ± 2.2	2.4	99.7 ± 3.0	3.0	

Three QC levels were used to study the recovery and matrix effect. As detailed in Table 1, the recovery of crizotinib and 2-Keto crizotinib in rat plasma were 94.2 to 100.5% and 92.5 to 93.5%, respectively. The calculated matrix effect were well within acceptable limits, ranging from 93.3% to 100.2% for crizotinib, and 98.4% to 102.7% for 2-Keto crizotinib, respectively. These results indicated that the matrix effect had minimal impact on analyte ionization and did not compromise the precision of the UPLC-MS/MS method.

The stability of crizotinib and 2-Keto crizotinib under various conditions are summarized in Table 2. Both analytes remained stable at room temperature for 3 h, 10 °C for 4 h, after three freeze-thaw cycles (−80 °C/RT), and during storage at −80 °C for 21 days, with the results within the acceptable error range (±15%).

Table 2 Stability results of crizotinib and 2-Keto crizotinib in rat plasma (n = 5).

Analytes	Concentration (ng/mL)	Room temperature (3 h)	10 °C (4 h)	Three freeze-thaw	21 days	
		RSD (%)	RE (%)	RSD (%)	RE (%)	RSD (%)	RE (%)	RSD (%)	RE (%)	
crizotinib	5	9.5	−9.0	3.3	0.5	10.9	−8.3	4.9	−1.9	
	120	4.7	−6.9	5.8	5.9	7.6	−7.1	3.5	4.7	
	240	4.3	−1.6	5.8	3.6	5.6	−7.4	5.8	6.9	
2-Keto crizotinib	2	2.3	9.9	4.7	2.2	1.8	11.9	7.0	14.7	
	40	3.4	2.0	0.6	0.6	1.9	−0.7	2.2	5.2	
	80	2.2	3.0	2.3	−0.6	1.5	−0.7	5.6	5.9	

Study on enzyme kinetics of crizotinib in vitro

As depicted in Figs. 3A–3B, the Km values of crizotinib were determined to be 1.00 µM in RLM and 15.00 µM in HLM, indicating that the enzyme in RLM exhibited a higher affinity for crizotinib compared to HLM. Figures 3C–3D illustrated the inhibitory effects of tropifexor on crizotinib metabolism, with IC50 values of 4.43 and 4.30 µM in RLM and HLM, respectively. This suggested that tropifexor exhibited moderate inhibition of crizotinib, with comparable potency in both microsomal systems.

Figure 3 Michaelis–Menten plots of crizotinib in RLM (A) and HLM (B), and IC50 values of tropifexor at various concentrations for crizotinib activity in RLM (C) and HLM (D).

The inhibition mechanisms were analyzed using the dissociation constants Ki and αKi, as presented in Fig. 4. Tropifexor was found to have a mixed mechanism of inhibition on crizotinib metabolism in both RLM and HLM. In RLM, tropifexor inhibited crizotinib metabolism through non-competitive and un-competitive mechanism, with Ki and αKi of 5.60 and 3.72 µM, respectively. In HLM, tropifexor inhibited crizotinib metabolism via competitive and non-competitive mechanism, with Ki and αKi of 4.93 and 13.38 µM, respectively.

Figure 4 Lineweaver-Burk double reciprocal plot, its secondary plot for αKi, and its secondary plot for Ki of tropifexor inhibiting crizotinib metabolism in RLM (A) and in HLM (B).

Data are presented as the means ± SD; n = 3.

Study of pharmacokinetic interaction in vivo

To assess the in vivo interaction between tropifexor and crizotinib, the rats were given 25 mg/kg crizotinib with or without tropifexor. Figure 5 shows the mean concentration–time curves of crizotinib and its metabolite 2-Keto crizotinib in rats, and Table 3 lists the key pharmacokinetic parameters derived from a non-compartmental model.

Figure 5 The mean plasma concentration–time curves of crizotinib (A) and its metabolite 2-Keto crizotinib (B) in the control group (crizotinib alone) and the experimental group (crizotinib with tropifexor) (n = 5).

Table 3 The pharmacokinetic parameters of crizotinib and 2-keto crizotinib in rats (n = 5, mean ± S.D.).

Compound	Crizotinib	2-Keto crizotinib	
Group	Crizotinib	Crizotinib + Tropifexor	Crizotinib	Crizotinib + Tropifexor	
AUC(0−t)(ng/mL*h)	4,098.16 ± 480.24	5,563.04 ± 1,112.56*	603.21 ± 97.71	877.14 ± 268.34	
AUC(0−∞) (ng/mL*h)	4,141.79 ± 490.95	5,669.41 ± 1,166.99*	604.85 ± 99.06	879.58 ± 268.53	
t1/2 (h)	6.83 ± 0.77	8.23 ± 1.65	4.82 ± 1.18	5.78 ± 1.79	
Tmax (h)	8.80 ± 1.79	8.80 ± 1.79	8.00 ± 0.00	8.80 ± 1.79	
CLz/F (L/h/kg)	6.10 ± 0.69	4.56 ± 0.90*	42.25 ± 7.06	30.44 ± 8.32*	
Cmax (ng/mL)	236.97 ± 45.30	325.92 ± 40.95*	51.05 ± 14.39	74.33 ± 26.83	
Notes.

Significant differences from control group, *P < 0.05.

Following a single oral dose of 25 mg/kg crizotinib, Cmax was reached at 8.80 ± 1.79 h with a value of 236.97 ± 45.30 ng/mL, and t1/2 was 6.83 ± 0.77 h. Compared to the control group, co-administration of tropifexor significantly increased AUC(0−t) and AUC(0−∞) of crizotinib by 35.7% and 36.9%, respectively, and Cmax by approximately 37.5%, while CLz/F was decreased by 25.2%. For the metabolite 2-Keto crizotinib, only CLz/F showed a 27.9% reduction, with no significant changes in other pharmacokinetic parameters. These findings suggested that tropifexor had the potential to inhibit the metabolism of crizotinib in vivo.

Discussion

Crizotinib has been widely utilized for the treatment of NSCLC due to its targeted action on specific genetic aberrations (Ou, 2011). The FDA approved crizotinib in 2011 for the treatment of locally advanced or metastatic ALK-positive NSCLC and later in 2016 for ROS1-positive metastatic NSCLC (Malik et al., 2014; Nadal et al., 2024). However, crizotinib induces Grade 1 or 2 liver abnormalities in 30% of patients, typically within two months of treatment, and these are reversible with dose reduction or interruption (Revol et al., 2020). Since crizotinib is primarily metabolized in the liver, liver function can significantly influence its elimination (El-Khoueiry et al., 2018).

Cancer patients often take multiple medications, increasing the likelihood of DDI (Firkins et al., 2018). In vitro studies using HLM and recombinant enzymes demonstrated that crizotinib is primarily metabolized by CYP3A, which significantly mediates the formation of both crizotinib lactam (PF-06260182) and O-dealkylation metabolites (Johnson et al., 2015). Previous studies have shown that co-administration of crizotinib with the strong CYP3A4 inhibitor ketoconazole resulted in a 2.2-fold increase in AUC compared to a single oral dose of 150 mg crizotinib (Xu et al., 2015). Additionally, a validated physiologically-based pharmacokinetic model predicting DDI between azole antifungals and crizotinib in cancer patients demonstrated that the AUC of crizotinib was increased by 84%, 58%, and 79% when co-administered with voriconazole, fluconazole, or itraconazole, respectively (Chen, Li & Chen, 2022). Clinically, careful monitoring of crizotinib plasma concentrations and potential interactions is essential when used with CYP3A inhibitors.

The previous study found that co-administration of tropifexor with the CYP3A4 inducer rifampin resulted in a 55% decrease in its Cmax, while co-administration with a CYP3A4 inhibitor led to only a 9% decrease in Cmax (Chen et al., 2022b), suggesting that tropifexor may compete for the CYP3A4 binding site and has the potential for DDI when used in combination with crizotinib.

This study evaluated the effect of tropifexor on crizotinib metabolism in RLM and HLM through both in vitro and in vivo experiments. In our experiment, the Km value in RLM was 1.00 µM, with a Vmax of 0.01457 pmol/min/µg protein, which is not similar to a previous study that reported the Km of crizotinib in RLM as 4.61 ± 0.28 µM (Wang et al., 2024). Further analysis from the IC50 and Ki values showed that tropifexor had a moderate inhibitory effect on crizotinib metabolism and exhibited a mixed inhibition mechanism. A similar drug interaction study found that proanthocyanidins also exhibited mixed inhibition (non-competitive and un-competitive) on crizotinib metabolism, which was consistent with the findings of this our study (Wang et al., 2024).

To further investigate the in vivo relevance of these findings, the study proceeded with animal experiments using rat model to assess the pharmacokinetics of crizotinib and its metabolite 2-Keto crizotinib, following co-administration with tropifexor. Based on the Cmax observed at 4 h after a single oral dose of 250 mg/d in humans (Ou, 2011), the dose of 25 mg/kg was chosen for rats by adjusting for the difference in body surface area. The results showed that tropifexor significantly altered the pharmacokinetic parameters of crizotinib, while the changes in 2-Keto crizotinib were not as pronounced. Under the same conditions in SD rats, after oral administration of crizotinib at a dose of 24 mg/kg, crizotinib was rapidly absorbed, reaching a mean Cmax of 309 ± 39.8 ng/mL at Tmax of 2.00 ± 0.00 h. The oral absolute bioavailability of crizotinib in rats was calculated to be 68.6 ± 9.63% (Qiu et al., 2016). In comparison, in this study, the single oral dose of 25 mg/kg resulted in a Cmax of 236.97 ± 45.30 ng/mL, which was consistent with the previous findings.

This study systematically evaluated the effect of tropifexor on the metabolism of crizotinib. In vitro experiments demonstrated that tropifexor inhibited crizotinib metabolism in both RLM and HLM, showing comparable IC50 values. Consistent with these findings, in vivo studies revealed that tropifexor suppressed the elimination of crizotinib and its primary metabolite 2-keto crizotinib, resulting in a marked increase in systemic exposure to crizotinib. Taken together, the concordance between in vitro and in vivo results suggested a potential DDI, whereby tropifexor may enhance crizotinib exposure and consequently increase the risk of adverse reactions. These results provided important evidence to support potential dose adjustment strategies when tropifexor is co-administered with crizotinib. Nevertheless, a limitation of this study is the reliance on SD rat model, which may not fully recapitulate human metabolic interactions. Future investigations should include clinical validation and further assessment of the long-term impact of this interaction on therapeutic efficacy and safety.

Conclusions

In summary, this study revealed that tropifexor significantly affected the pharmacokinetics of crizotinib in rats, increasing its AUC(0−t), AUC(0−∞) and Cmax while reducing CLz/F, which suggested an inhibitory effect on crizotinib metabolism. The unchanged pharmacokinetics of 2-Keto crizotinib, except for reduced clearance, highlighted the selective impact on crizotinib. In vitro studies indicated mixed inhibition mechanisms in both RLM and HLM. These findings suggested potential DDI between tropifexor and crizotinib, emphasizing the need for further clinical studies to evaluate their therapeutic implications and underlying mechanisms.

Supplemental Information

Supplemental Information 1 Raw data

Supplemental Information 2 The ARRIVE guidelines 2.0: author checklist

We are thankful for the help and support of Ren-ai Xu.

Additional Information and Declarations

Competing Interests

Author Contributions

Ethics

Data Availability

The authors declare there are no competing interests.

An Shen conceived and designed the experiments, authored or reviewed drafts of the article, and approved the final draft.

Hailun Xia performed the experiments, prepared figures and/or tables, and approved the final draft.

Jun Wu analyzed the data, prepared figures and/or tables, and approved the final draft.

Liang Tao conceived and designed the experiments, authored or reviewed drafts of the article, and approved the final draft.

Jie Chen performed the experiments, prepared figures and/or tables, and approved the final draft.

HangJuan Lin conceived and designed the experiments, analyzed the data, authored or reviewed drafts of the article, and approved the final draft.

The following information was supplied relating to ethical approvals (i.e., approving body and any reference numbers):

The study protocol followed the ARRIVE guidelines and was approved by the Institutional Animal Care and Use Committee of the First Affiliated Hospital of Wenzhou Medical University (Ethics approval number: WYYY-IACUC-AEC-2024-079).

The following information was supplied regarding data availability:

The raw data is available in the Supplementary File.

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
