# Peer review of "Interaction between crizotinib and tropifexor through in vitro and in vivo studies"

_PeerJ, doi:10.7717/peerj.20256_

## Round 0.1 · original submission · Major Revisions

**Language Note:** When preparing your next revision, please ensure that your manuscript is reviewed either by a colleague who is proficient in English and familiar with the subject matter, or by a professional editing service. PeerJ offers language editing services; if you are interested, you may contact us at [email protected] for pricing details. Kindly include your manuscript number and title in your inquiry. – PeerJ Staff

Reviewer 1 ·

Basic reporting

Dear authors,
Your work is interesting as a researcher in this domain. I highly recommend having such studies. However, I found some suggestions to improve your manuscript. Below are the comments:

1. Extend the Introduction by adding a Literature survey paragraph, and at the end of the Introduction, write 4-5 lines about the methods, what, and why it was done while keeping the objectives in mind. You may refer to any article for this, and you get the idea.

2. Authors are recommended to read recent lung cancer studies: Relevant topics include: the application of machine learning techniques in identifying therapeutic targets and predicting treatment outcomes; the use of multisampling-based molecular docking and simulation methods to discover multitargeted inhibitors; and structural modeling approaches for identifying compounds that inhibit key enzymes such as CDKs and transferase kinases in lung cancer.

3. Include 1-2 lines at the start of the Introduction about the mortality and how the WHO treated it as a threat, etc.

4. There are too many tables. Can you merge them by any means into one table, so that we can have an idea of multiple tables, or maybe something like this?

Experimental design

-

Validity of the findings

-

Reviewer 2 ·

Basic reporting

The study design and execution were good, and the manuscript framework was good, but it has a few clarifications needed or mistakes to be corrected. Here are my comments to improve the manuscript.

Experimental design

In the section In vitro assay, "positive control inhibitors are not used as an inhibitor, if I am not wrong.
Why were the rats pre-treated for 30 min with tropifexor prior to dose administration?

Validity of the findings

-

·

Basic reporting

1. Start the abstract with a clear statement on the scope, relevance, and intention of the study, before describing the main results.

2. End the abstract with a clear statement about the main conclusions and perspectives of the work.

3. Justify the chosen concentrations of tropifexor and crizotinib in inhibition studies (e.g., relevance to clinical exposure).

4. The manuscript needs to be grammatically correct; there are many errors.

Experimental design

1. Provide purity information for key chemicals (e.g., crizotinib, tropifexor, alectinib) and briefly justify the choice of alectinib as an internal standard.

2. Add missing UPLC-MS/MS method details, including:
a) Flow rate of the mobile phase.
b) MRM transitions (precursor and product ions, collision energy) for each analyte and internal standard.
c) Justification for the gradient elution program.

3. For enzyme incubation experiments, specify whether RLM and HLM were pooled or from single donors and why 30 minutes was chosen as the incubation time (e.g., linearity of metabolite formation).

4. Include more details on sample collection: specify the heparin concentration used as an anticoagulant and whether hemolyzed samples were excluded.

Validity of the findings

1. In the in vivo pharmacokinetic study:
a) Justify the dose of tropifexor (20 µg/kg) and crizotinib (25 mg/kg).
b) Explain how animals were randomized, whether blinding was performed, and provide reasoning for the sample size (n=5/group).

2. Regression equations for calibration curves are unclear because the format (0.004216*x ± 0.001441) is ambiguous. Clarify whether ± values represent the standard deviation of slope/intercept or confidence intervals.

3. In the method validation, recovery and matrix effect results are summarized, but ranges (e.g., “94.2 100.5%”) are not properly formatted (likely missing “–” or “to”). Please correct these formatting errors.

Additional comments

1. The first or final paragraph of the discussion should clearly describe the main conclusions of the work, their importance, and potential for further studies.

---

## Round 0.2 · accepted · Accept

Thank you for revising your manuscript to address the reviewers' concerns. Reviewers 1 and 3 now recommend acceptance and I am satisfied that the comments of Reviewer 2 have been addressed. The manuscript is now ready for publication.

Reviewer 1 ·

Basic reporting

Now the manuscript is good to publish.

Experimental design

Now the manuscript is good to publish.

Validity of the findings

Now the manuscript is good to publish.

Additional comments

Now the manuscript is good to publish.

·

Basic reporting

The manuscript is well-prepared, methodologically sound, and addresses a relevant pharmacological question regarding drug–drug interactions between crizotinib and tropifexor.

Experimental design

The experimental design is rigorous, the methods are described with sufficient detail for reproducibility.

Validity of the findings

The results are clearly presented and statistically supported.

Additional comments

I did not find any major flaws in study design, data interpretation, or ethical compliance.